# [18F]FDG PET/CT: Lung Nodule Evaluation in Patients Affected by Renal Cell Carcinoma

Lighea Simona Airò Farulla [1,2,†], Laura Lavinia Travaini [1,†], Mariarosaria Cuomo [1,2], Domenico Galetta [2,3], Francesco Mattana [1], Samuele Frassoni [4], Giuseppe Buonsanti [1], Lorenzo Muraglia [1], Giulia Anna Zuccotti [1,2], Vincenzo Bagnardi [4], Lorenzo Spaggiari [2,3] and Francesco Ceci [1,2,*]

1 Division of Nuclear Medicine, IEO European Institute of Oncology, IRCCS, 20141 Milan, Italy
2 Department of Oncology and Hemato-Oncology, University of Milan, 20141 Milan, Italy
3 Division of Thoracic Surgery, IEO European Institute of Oncology, IRCCS, 20141 Milan, Italy
4 Department of Statistics and Quantitative Methods, University of Milan-Bicocca, 20141 Milan, Italy
* Correspondence: francesco.ceci@ieo.it
† These authors contributed equally to this work.

**Abstract:** Renal Cell Carcinoma (RCC) is generally characterized by low-FDG avidity, and [18F]FDG-PET/CT is not recommended to stage the primary tumor. However, its role to assess metastases is still unclear. The aim of this study was to evaluate the diagnostic accuracy of [18F]FDG-PET/CT in correctly identifying RCC lung metastases using histology as the standard of truth. The records of 350 patients affected by RCC were retrospectively analyzed. The inclusion criteria were: (a) biopsy- or histologically proven RCC; (b) Computed Tomography (CT) evidence of at least one lung nodule; (c) [18F]FDG-PET/CT performed prior to lung surgery; (d) lung surgery with histological analysis of surgical specimens; (e) complete follow-up available. A per-lesion analysis was performed, and diagnostic accuracy was reported as sensitivity and specificity, using histology as the standard of truth. [18F]FDG-PET/CT semiquantitative parameters (Standardized Uptake Value [SUVmax], Metabolic Tumor Volume [MTV] and Total Lesion Glycolysis [TLG]) were collected for each lesion. Sixty-seven patients with a total of 107 lesions were included: lung metastases from RCC were detected in 57 cases (53.3%), while 50 lesions (46.7%) were related to other lung malignancies. Applying a cut-off of SUVmax ≥ 2, the sensitivity and the specificity of [18F]FDG-PET/CT in detecting RCC lung metastases were 33.3% (95% CI: 21.4–47.1%) and 26% (95%CI: 14.6–40.3%), respectively. Although the analysis demonstrated a suboptimal diagnostic accuracy of [18F]FDG-PET/CT in discriminating between lung metastases from RCC and other malignancies, a semiquantitative analysis that also includes volumetric parameters (MTV and TLG) could support the correct interpretation of [18F]FDG-PET/CT images.

**Keywords:** PET; lung nodule; renal cell carcinoma; FDG

## 1. Introduction

Renal cell carcinoma (RCC) accounts for around 3% of all cancers and 80–85% of primary renal neoplasms [1]. With higher incidence rates in North America, Australia and Europe, the global age-standardized incidence rate is 4 per 100,000 people per year, while lower incidence rates have been found in Japan, Africa, India and China [2]. The estimated average 5-year survival rates for patients with RCC are 96% for those presenting with stage II disease, 64% for those with stage III disease and 23% for those with stage IV disease [3]. The five-year survival rate has significantly improved in recent years compared with the past in patients with RCC. In particular, this improvement has been observed especially for those tumors smaller than 2 cm and for patients with localized or regional disease compared with those with distant metastasis [2].

The World Health Organization identified three histological types of RCC: clear cell RCC (constituting 80–90% of the cases), papillary RCC (10–15% of the cases) and chromophobe RCC (4–5% of the cases) [1,2]. Additional subtypes were then identified, based on histological features such as different nuclear size (larger or smaller), irregularity and nucleolar prominence, defined as nuclear grades, which, at first, were found to be more predictive of the development of distant metastases after nephrectomy than the pathologic stage, the cell arrangement, the cell type or the tumor size [2].

In general, 20–40% of localized RCC develop metastases late, with 5-year patient survival below 10% [4], and the most common sites of recurrences are the lung, regional lymph nodes and bone; this justifies the need of surveillance, as it has been shown that a subset of these patients may benefit from therapies for recurrent disease [5]. This is mainly because in the past 15 years, the treatment of metastatic RCC has significantly changed and requires, in most cases, cytoreductive nephrectomy before starting systemic drugs, which include immunotherapy drugs, mTOR inhibitors and antiangiogenic agents [5]. Lung metastases from RCC are usually small (0.5 to 2 cm diameter), well-defined round or ovoid nodules, solitary or multiple ("cannonball" metastases) and asymptomatic [6]. In most cases, the diagnosis of lung metastasis is made by chest Computed Tomography (CT), which allows an early diagnosis in the absence of patient symptoms, while only a very small percentage of patients have been diagnosed with metastasis because of symptoms such as dyspnea, cough, pleuritic chest pain, hemoptysis or weight loss [7]. A follow-up program after surgery for localized RCC in particular has been recommended, based on possible treatment options in the case of recurrence [1]. The intervals between CT scans of the chest and abdomen differ according to patients' risk factors: for those at low risk, an annual CT scan may be enough, while for those at high risk, the intervals are reduced to 3–6 months in the first two years after surgery [1].

Fluorodeoxyglucose ([18F]FDG) is the predominant Positron Emission Tomography (PET) tracer broadly used in oncology. Its uptake depends on the cancer's ability to use the membrane glucose transporter GLUT1, that is usually upregulated in cancerous cells (with increased levels of intracellular hexokinase and low levels of glucose-6-phosphatase), or other overexpressed enzymes such as lactate dehydrogenase (LDH) [8]. When GLUT-1 actively transports FDG into the cell, hexokinase converts it to FDG-6-phosphate, which is not one of the substrates for the other steps of glycolysis and is trapped in the cell itself [8]. The accumulation of FDG-6-phosphate is thus directly proportional to the glucose metabolic activity of the cell, and it is possible to measure this accumulation semi-quantitatively with the standardized uptake value (SUV) [9]. This glucose analogue is then labeled with 18F, which is a fluorine radioisotope produced by the cyclotron and has a short half-life (approximately 109.7 min) [10]. PET is a tomographic technique that allows the noninvasive quantitative assessment of functional and biochemical processes by measuring the three-dimensional distribution of positron-emitting radiopharmaceuticals [10].

RCC is not typically evaluated with [18F]FDG-PET/CT due to the physiologic renal excretion of the tracer used, which might limit its diagnostic accuracy, while the reasons for the inconsistent nature of FDG uptake in RCC remain unknown. [11]. As a result, [18F]FDG-PET/CT is not routinely recommended as an imaging tool in this setting by clinical guidelines such as AUA, ESMO and EAU [1,5,11]. However, [18F]FDG-PET/CT can be proposed for those patients presenting a high likelihood of metastatic disease, including lung metastases, when molecular imaging is suggested in addition to CT, partly because it has been found that PET appears to be more sensitive for the detection of metastatic RCC lesions than for the detection of primary renal tumors [11].

Therefore, the aim of this study was to evaluate the diagnostic accuracy of [18F]FDG-PET/CT in correctly identifying RCC lung metastases in a cohort of patients with diagnostic suspicion of lung nodules, using histopathological analysis of the surgical specimens as the standard of truth.

## 2. Materials and Methods

### 2.1. Objectives

The primary endpoint of this study was to assess the diagnostic accuracy of [$^{18}$F]FDG-PET/CT in assessing the presence of lung metastases from RCC, in terms of both sensitivity and specificity.

The secondary endpoint of this study was to evaluate the correlation between [$^{18}$F]FDG-PET/CT semi-quantitative parameters (SUVmax, Metabolic Tumor Volume [MTV] and Total Lesion Glycolysis [TLG]) with tumor histology (RCC metastases vs. other malignancies).

### 2.2. Study Design and Patients Selection

This was a retrospective, single-center, single-arm, open-label study in RCC patients treated and followed-up at our institution. The clinical records of patients from February 2004 to October 2020 (study cut-off date) were evaluated.

The inclusion criteria were: (1) biopsy or histologically proven RCC; (2) suspicion of lung metastasis (at least one lung nodule observed in contrast-enhanced CT (ceCT); (3) [$^{18}$F]FDG-PET/CT as the baseline procedure, prior to lung surgery; (4) lung surgery with subsequent histopathological analysis of surgical specimens; (5) availability of all clinical, pathology and imaging data (no lost follow-up). The exclusion criteria were: (1) low-quality PET images (e.g., wrong input of the PET parameters, glycemia > 200 mg/dL, para-vein injection, 2D scanners); (2) proven metastatic disease other than in the lung. The study was approved by the local ethical committee and institutional scientific review board (IEO Trial-ID: 3490).

### 2.3. [$^{18}$F]FDG-PET/CT

[$^{18}$F]FDG-PET/CT was performed with a standard procedure according to the European Association of Nuclear Medicine (EANM) procedural guidelines [10]. PET images were acquired on a PET/CT scanner (GE DMI-DDR, GE Healthcare, Milwaukee, WI, USA), 60 min after the intravenous injection of the radiopharmaceutical (as suggested by the current guidelines to achieve the optimal biodistribution of the tracer). An activity of 3 MBq/kg was injected (median 262 MBq; range 171–388 MBq). SUV/body-weighted (SUV/bw) was defined as the ratio of activity per unit volume of a region of interest (ROI) to the activity per unit of whole body volume (SUV = activity concentration/[injected dose/body weight]), calculated on attenuation-corrected images.

### 2.4. Image Analysis

First, the [$^{18}$F]FDG-PET/CT images were reviewed by three nuclear medicine physicians (LLT, MC, LSAF) and interpreted with consensus (rule 2:1) as positive or negative according to a qualitative evaluation. Subsequently, each [$^{18}$F]FDG-PET/CT image was re-examined, and the positivity of the [$^{18}$F]FDG-PET/CT was determined on the basis of an arbitrary cut-off of SUVmax $\geq 2$ determined from data in the literature.

Semi-quantitative analysis was performed using an AW Server Workstation 2.0 (GE Medical Systems, Milwaukee, WI, USA) providing multiplanar reformatted images. The three nuclear medicine physicians identified all the known lung nodules on CT and calculated, on corresponding PET images, the semiquantitative parameters SUVmax, MTV and TLG, by setting a spherical volume of interest (VOI) over the regions of interest.

### 2.5. Statistical Analysis

Continuous data are reported as median and interquartile ranges (IQR). Categorical data are reported as counts and percentages.

The accuracy in defining a lesion as an RCC lung metastasis, according to both the PET/CT lung result and SUVmax, was evaluated using the histopathological result as the standard of truth. Sensitivity (SE) and specificity (SP), with their 95% confidence intervals (95% CI), were calculated for the two methods. The Wilcoxon rank-sum test was performed to evaluate the distribution of the [$^{18}$F]FDG-PET/CT semi-quantitative body parameters

(SUVmax, MTV and TLG) among the lesions with and without RCC lung metastases. A *p*-value less than 0.05 was considered statistically significant.

All analyses were performed with the statistical software SAS 9.4 (SAS Institute, Cary, NC, USA).

## 3. Results

### 3.1. Patient Population

Overall, the clinical records of 350 patients affected by RCC and with CT evidence of at least one lung nodule and referred to out Institute were retrospectively analyzed. After applying the inclusion/exclusion criteria, 70 patients presenting with 111 lung nodules were initially included in the analysis. The median diameter of the 111 lung nodules was 12 mm (IQR 8.5–18.5 mm). Three patients were subsequently excluded from this population since they showed non-malignant nodules at the histopathological analysis. Accordingly, 67 (67/70; 96%) patients presenting 107 lung nodules were considered as the patient cohort eligible for primary endpoint analysis (Figure 1). The median diameter of the 107 lung nodules was 12 mm (IQR 8–18.5 mm). The population characteristics are summarized in Table 1. Almost all patients (65/67) underwent surgery as primary therapy. Total nephrectomy was performed in most cases (79%), partial nephrectomy in 6% of the cases, and simple resection/enucleation in 12% of them. Two patients did not undergo surgery due to clinical decision.

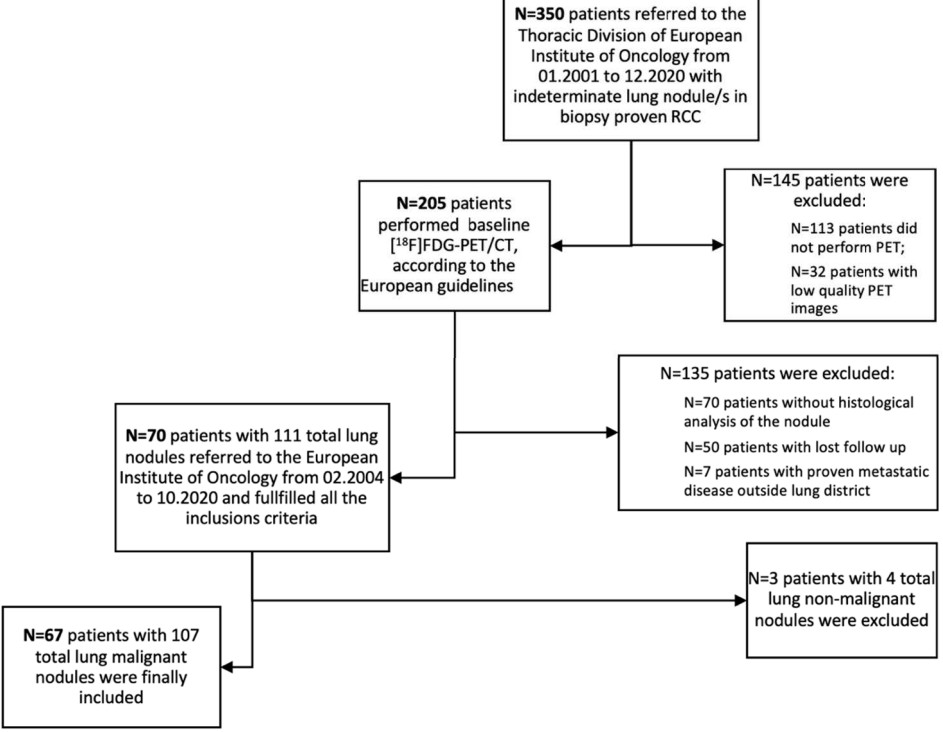

**Figure 1.** Flowchart for the study population [10].

ceCT was performed in all patients prior to PET imaging. Twenty-five patients (37%) had more than one lung nodule in ce-CT, while forty-two patients (63%) had a solitary lung nodule. All patients underwent lung surgery with histological analysis of the surgical specimens. Fifty-two percent (52%) of the patients (35/67) underwent wedge resection, 43% (29/67) underwent lobectomy, and 4% (3/67) had pneumonectomy. Histological analysis of the 107 malignant nodules revealed lung metastases from RCC in 53% of the patients and primary adenocarcinoma in 33% of them. Other less frequent malignant lesions were squamous carcinoma (7%) and other tumors (7%, such as large cell neuroendocrine carcinoma, large cell undifferentiated carcinoma, lung carcinosarcoma and lymphoma).

**Table 1.** Demographic, tumor and treatment characteristics of the study population (*n* = 67).

| Variable | Level | Overall (*n* = 67) |
|---|---|---|
| Age at the date of PET (y), median (IQR) | | 66 (58–73) |
| Gender, *n* (%) | Female | 19 (28.4) |
| | Male | 48 (71.6) |
| Type of primary therapy, *n* (%) | No surgery | 2 (3.0) |
| | Total nephrectomy | 53 (79.1) |
| | Partial nephrectomy | 4 (6.0) |
| | Enucleation | 8 (11.9) |
| Type of lung surgery, *n* (%) | Wedge resection | 35 (52.2) |
| | Lobectomy | 29 (43.3) |
| | Pneumonectomy | 3 (4.5) |
| Number of lesions, *n* (%) | 1 | 42 (62.7) |
| | 2 | 16 (23.9) |
| | 3 | 5 (7.5) |
| | 4 | 2 (3.0) |
| | 5 | 2 (3.0) |
| Lung histology, *n* (%) * | RCC metastasis | 57 (53.3) |
| | Adenocarcinoma | 35 (32.7) |
| | Squamous carcinoma | 7 (6.5) |
| | Other tumors | 8 (7.5) |
| Sum of the lesions size (mm), median (IQR) | | 19 (12–33) |

* *n* and % refer to the 107 lesions observed in the 67 patients included in the analysis.

All patients included were treated and followed-up according to the best standard-of-care clinical practice at our institution and according to international procedural guidelines. All decisions were taken in a multi-disciplinary tumor board setting.

*3.2. PET Parameters*

In the primary endpoint analysis, the 107 nodules analyzed were sorted into two subgroups based on the histological results post-lung surgery: one subgroup included lung metastases from RCC (53%), while the other subgroup included all other malignant nodules (47%). A first analysis was performed considering the qualitative analysis of [$^{18}$F]FDG-PET/CT images, yielding a sensitivity of 49.1% (95% CI: 35.6–62.7%) and a specificity of 10.0% (95% CI: 3.3–21.8%). When applying the SUVmax $\geq$ 2 arbitrary cut-off to consider a lesions as an RCC lung metastasis, a sensitivity of 33.3% (95% CI: 21.4–47.1%) and specificity of 26% (95% CI: 14.6–40.3%) were observed (Table 2).

**Table 2.** Lesion classification and diagnostic performance of the PET/CT lung results and lung SUV max, with the histopathological result as the gold standard to define a lesion as an RCC lung metastasis (*n* = 107).

| | Histopathological Result | | | Diagnostic Performance | |
|---|---|---|---|---|---|
| | **No RCC Lung Met** | **RCC Lung Met** | **Total** | **SE (95% CI)** | **SP (95% CI)** |
| PET/CT qualitative analysis | | | | 49.1% | 10.0% |
| Negative | 5 | 29 | 34 | (35.6–62.7%) | (3.3–21.8%) |
| Positive | 45 | 28 | 73 | | |
| Lung SUV max | | | | 33.3% | 26.0% |
| <2 (Negative) | 13 | 38 | 51 | (21.4–47.1%) | (14.6–40.3%) |
| ≥2 (Positive) | 37 | 19 | 56 | | |
| Total | 50 | 57 | 107 | | |

The median (IQR) of SUVmax was 1.58 (0.80–3.32) for the RCC lung metastases and 4.79 (1.96–10.5) for the other malignancies (*p* < 0.001). In all lesions, the semiquantitative

volumetric parameters (MTV and TLG) were also calculated. MTV and TLG were significantly lower in RCC metastases compared to other lung malignancies (Table 3 and Figure 2A,C).

**Table 3.** Distribution of FDG-PET/CT semi-quantitative body parameters among lesions with and without metastases (*n* = 107).

| Variable | Histopathological Result | | *p*-Value |
|---|---|---|---|
| | No Metastasis (*n* = 50) | Metastasis (*n* = 57) | |
| Lung SUVmax, median (IQR) | 4.79 (1.96–10.5) | 1.58 (0.80–3.32) | <0.001 |
| Lung MTV (cm$^3$), median (IQR) | 1.49 (0.49–4.60) | 0.12 (0.00–1.47) | <0.001 |
| Lung TLG, median (IQR) | 5.64 (1.35–16.2) | 0.05 (0.00–3.25) | <0.001 |

A secondary analysis was then performed considering only lung lesions equal to or less than 10 mm in CT or ceCT (46 out of 107 nodules) and subdividing them into the same two histologic subgroups mentioned above: lung metastases from RCC (36/46; 78%) and other lung malignancies (10/46; 22%). We then compared these histological subgroups of lung nodules equal or less than 10 mm with lung findings from the original [$^{18}$F]FDG-PET/CT reports, yielding a sensitivity and a specificity of 30.6% (95% CI: 16.4–48.1%) and 20% (95% CI: 2.5–55.6%), respectively, and with SUVmax ≥ 2 as the cut-off discriminator, obtaining a sensitivity of 11.1% (95% CI: 3.1–26.1%) and a specificity of 50% (95% CI: 18.7–81.3%) (Table 4).

**Table 4.** Lesion classification and diagnostic performance of the PET/CT lung results and lung SUV max, with the histopathological result as the gold standard to define a lesion as an RCC lung metastasis, among lesions with size ≤ 10 mm (*n* = 46).

| | Histopathological Result | | | Diagnostic Performance | |
|---|---|---|---|---|---|
| | No RCC Lung Met | RCC Lung Met | Total | SE (95% CI) | SP (95% CI) |
| PET/CT lung result | | | | 30.6% (16.4–48.1%) | 20.0% (2.5–55.6%) |
| Negative | 2 | 25 | 27 | | |
| Positive | 8 | 11 | 19 | | |
| Lung SUV max | | | | 11.1% (3.1–26.1%) | 50.0% (18.7–81.3%) |
| <2 (Negative) | 5 | 32 | 37 | | |
| ≥2 (Positive) | 5 | 4 | 9 | | |
| Total | 10 | 36 | 46 | | |

Two clinical exemples are reported in Figures 3 and 4.

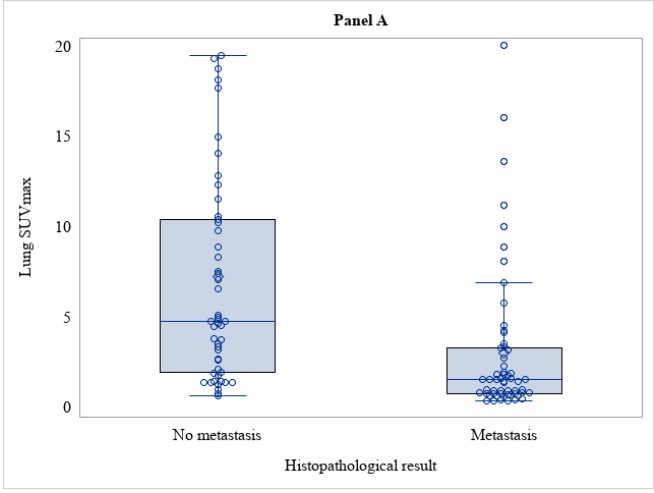

**Figure 2.** *Cont.*

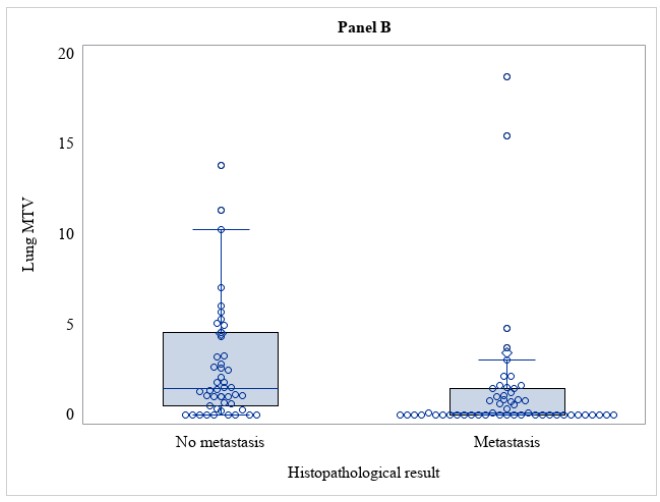

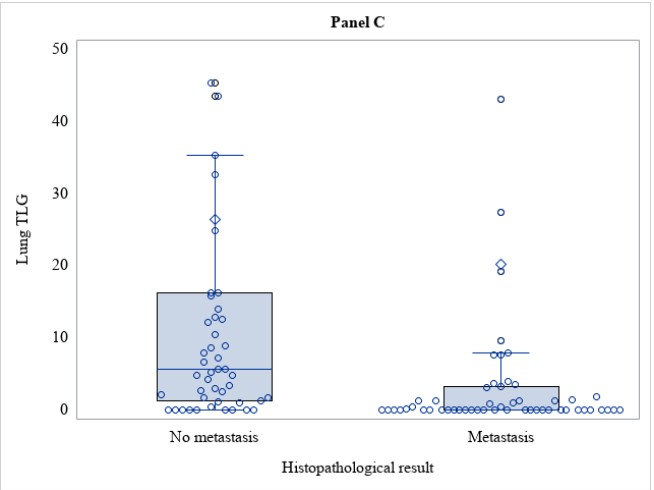

**Figure 2.** Distribution of SUVmax (Panel (**A**)), MTV (Panel (**B**)), TLG (Panel (**C**)) among lesions with and without metastasis (*n* = 107).

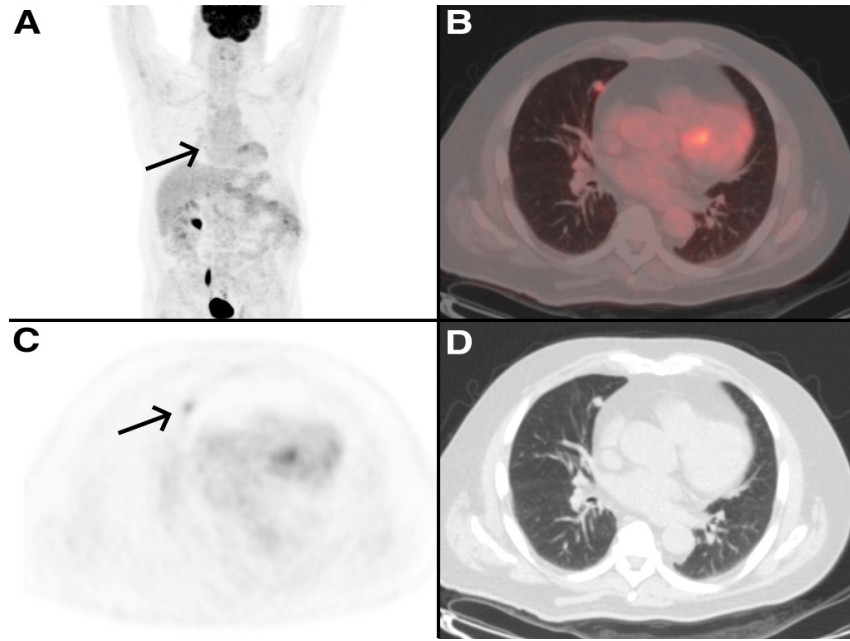

**Figure 3.** A 72-year-old man treated 3 years earlier with total nephrectomy of the left kidney for RCC,

referred to medical consultation for dyspnea. A ceCT scan was performed showing a lung nodule in the right middle lobe. [18F]FDG PET/CT was also performed and showed a mild uptake of the tracer in the lung nodule (SUVmax 2.8, arrows). Five months later, he underwent wedge resection with histological examination of the surgical specimens, which confirmed that the lung nodule was a metastasis from the RCC. (**A**) Maximum Intensity Projection (MIP) image of [18F]FDG PET/CT (**B–D**) PET/CT fusion image, PET and CT images of [18F]FDG PET/CT.

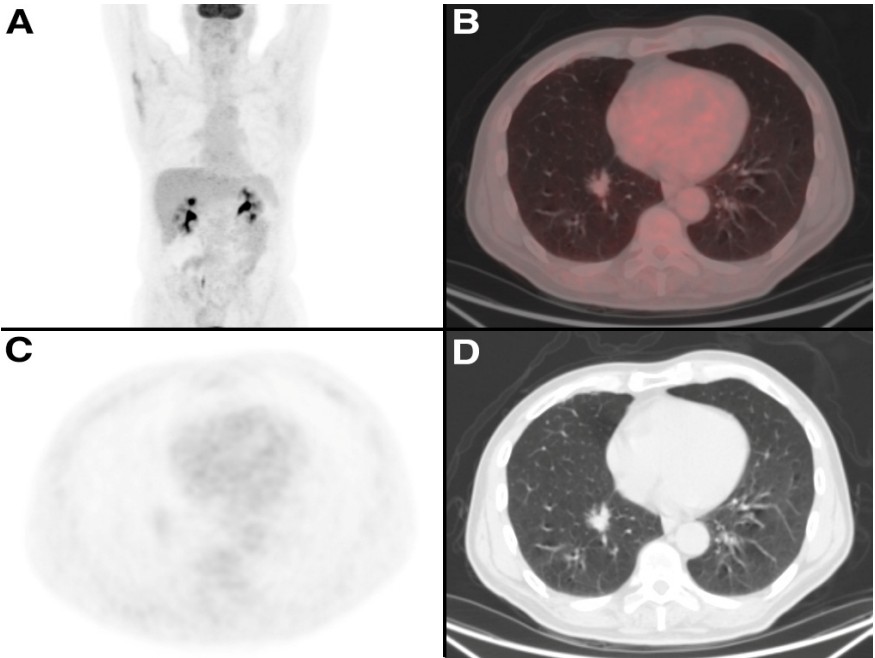

**Figure 4.** A 78-year-old man treated 7 years earlier with enucleation for RCC, underwent a ceCT scan which showed an irregular lung mass in the right lower lobe. [18F]FDG PET/CT was also performed and showed no uptake of the tracer in the lung nodule. Two months later, he underwent lobectomy with histological examination of the surgical specimens, which confirmed that the lung nodule was an adenocarcinoma. (**A**) Maximum Intensity Projection (MIP) image of [18F]FDG PET/CT (**B–D**) PET/CT fusion image, PET and CT images of [18F]FDG PET/CT.

## 4. Discussion

Chest ceCT scan is strongly recommended in patients with RCC (except cT1a), firstly because it is considered the most accurate method to identify the presence of lung metastasis, and secondly because the lung is one of the most common sites of metastasis from RCC. In addition, it is well known that patients who have only lung metastasis from RCC have a better prognosis than those who also have metastasis to other organs (especially bone and liver). Furthermore, the correct identification of lung metastasis is crucial, as surgical resection can be performed as soon as the lung nodule is identified [5,12,13]. As already mentioned, the potential role of [18F]FDG-PET/CT in the evaluation of recurrent RCC rather than in tumor staging in this context deserves to be explored. [18F]FDG-PET/CT is a one-stop-shop imaging procedure able to give a correct evaluation of the real burden of the disease for many solid malignancies. However, its role in RCC is still debated due to the low FDG avidity exhibited by this specific tumor histotype. Some recent studies emphasize the quality of [18F]FDG-PET/CT in differentiating solitary pulmonary nodules when the chest ceCT results are indeterminate. In this regard, some authors who performed three sequential scans in 43 patients with metastatic RCC found that the most common metabolically active metastatic sites were precisely the lungs and the lymph nodes [14]. The findings in the literature seem promising, as the diagnostic accuracy of [18F]FDG-PET/CT in these studies ranged from 87% to 92%, and many of them also highlighted that [18F]FDG-PET/CT can also show lesions not visible on ceCT [15,16].

However, in our study, less promising results were observed, as well as a very low diagnostic accuracy when comparing the qualitative analysis of [<sup>18</sup>F]FDG-PET/CT with the effective histology of the nodule. The results were slightly better when considering an arbitrary cut-off of SUVmax $\geq 2$. SUVmax, indeed, showed a statistically significant lower value in lung metastases from RCC than in other lung malignancies, as expected since RCC is a disease with low glucose metabolism. Although no cut-off value of SUVmax was found to provide an adequate diagnostic accuracy, SUVmax was still higher than the background in RCC lung metastases, resulting in the greater significance of the volumetric semiquantitative parameters (MTV and TLG), which were shown to be significantly different.

In our analysis, the standard of truth was histology rather than another diagnostic procedure, such as ceCT or clinical follow-up. This methodology is different compared to others published at present in the literature, as reported in a study that compared the accuracy of [<sup>18</sup>F]FDG-PET/CT with chest CT, showing a sensitivity of 75.0% and a specificity of 97.1% for lung metastases detected with the former technique compared with 91.1% and 73.1%, respectively, obtained with chest CT [11]; similar results were indeed obtained from another study that used, as a comparator, follow-up data obtained by conventional imaging (CT/MRI) and [<sup>18</sup>F]FDG-PET-CT, and histopathological confirmation only when possible, achieving a sensitivity of 80.6% with [<sup>18</sup>F]FDG PET/CT compared with that of 100% obtained with ceCT [16]. However, the authors reported that the nodules truly positive on CT and missed by [<sup>18</sup>F]FDG-PET/CT had a small diameter (less than 12 mm [11,17]). The exclusion of small nodules might result in an overestimation of the diagnostic accuracy of [<sup>18</sup>F]FDG-PET/CT because lung metastases from RCC present generally small dimensions. Lung nodules greater than 10 mm are well studied with CT due to their morphologic features and changes over time, which are frequently sufficient to identify RCC metastases over other lung malignancies, whereas distinguishing nonspecific lesions from small metastatic nodules with CT is more challenging [18]. Moreover, biopsy analysis or histological characterization is clinically indicated for lung nodules greater than 10 mm, and thus [<sup>18</sup>F]FDG-PET/CT positivity or negativity assumes limited significance, even if it has been observed that [<sup>18</sup>F]FDG-PET/CT may have an unfavorable prognostic value in the case of a positive scan [18,19]. All the patients in our cohort, including patients with nodules smaller than 10 mm underwent surgery. However, as also noted in Figure 1, many of the patients who underwent [<sup>18</sup>F]FDG-PET/CT did not then have surgery (70/205; 34%); therefore, in a real-world clinical scenario, it is not true that all patients with a history of RCC and a CT finding of a lung nodule underwent surgery, and this is especially true for patients with small nodules.

On the basis of these considerations (although it may appear unusual), we carried out a second analysis taking into account only small nodules (<10 mm), for which a surgical approach is not usually indicated and which are generally excluded from other studies in the literature, and showed that, even for this sample of our cohort, a poor diagnostic accuracy was detected; this result is, by the way, quite similar to those obtained by considering also larg nodules. The diagnostic accuracy, however, was better when a qualitative analysis of [<sup>18</sup>F]FDG-PET/CT was performed than when the arbitrary cut-off of SUVmax $\geq 2$ was used.

Finally, we thought it would be useful to make a third analysis using both the qualitative analysis of [<sup>18</sup>F]FDG-PET-CT and the arbitrary cut-off of SUVmax $\geq 2$ to discriminate malignant nodules (107/111) from non-malignant nodules (4/111); this analysis could show a much better diagnostic accuracy.

Definitely, one of the limitations of this last analysis is that a non-malignant nodule sample is quite small compared with a malignant one, but this is a limitation caused by the obvious selection bias of this study, as all 70 patients had a history of RCC and thus were cancer patients; it is also a selection bias due to the fact that though all our patients received histological confirmation, it is clear that a patient with a nodule that was frankly benign did not undergo surgery and thus subsequent histological characterization.

One of the biggest strengths of this study, however, remains that it provides a real-world scenario, which is even more accurate when we consider that our institute is a tertiary cancer center and, therefore, it is indeed a single center but still a reference center.

A final consideration is that the diagnostic accuracy improved by considering the results of the [$^{18}$F]FDG-PET-CT reports rather than the arbitrary cutoff of SUVmax $\geq 2$. This might indicate that, in such nuanced cases, the nuclear physician's expertise has greater accuracy than the semi-quantitative PET parameters.

## 5. Conclusions

In accordance with these data, obtained from a histology-based validated cohort, the use of [$^{18}$F]FDG-PET/CT to correctly identify lung metastases from RCC should be discouraged, as our analysis showed a suboptimal diagnostic accuracy of [$^{18}$F]FDG-PET/CT in discriminating between lung metastases from RCC and other lung malignancies. It is necessary to point out, however, that metastatic patients in whom [$^{18}$F]FDG-PET/CT has been shown to have added value, including prognostic value, were excluded from this analysis. Therefore, the take-home message is that, when [$^{18}$F]FDG-PET/CT is used in the restaging setting of metastatic RCC patients, it is necessary to keep in mind that metastases from RCC have a low SUVmax value (however, generally higher than the background), and it is important not to rely only on a qualitative analysis but to perform also a semiquantitative analysis using volumetric parameters (MTV and TLG) in addition to SUVmax, because they can support image interpretation.

Further perspective multi-center studies with a larger study case number are required to establish the diagnostic value of [$^{18}$F]FDG-PET/CT imaging for RCC patients with lung nodules. A final consideration is that in recent years, the role of other radiotracers (e.g., [$^{89}$Zr]Girentuximab or radiolabeled FAP inhibitors) in the detection of RCC metastases has been emerging; as these tracers may have greater diagnostic accuracy than [$^{18}$F]FDG-PET/CT, their role in this setting should be investigated with further studies.

**Author Contributions:** Conceptualization, L.S.A.F., L.L.T. and F.C.; methodology, F.M., L.M. and F.C.; formal analysis, S.F. and V.B.; investigation, L.S.A.F., L.L.T., M.C. and G.B.; resources, D.G., V.B. and L.S.; data curation, S.F. and V.B.; writing—original draft preparation, L.S.A.F., L.L.T. and F.C.; writing—review and editing, F.M., L.M., G.A.Z. and F.C.; visualization, L.S.A.F., S.F. and V.B.; supervision, L.L.T., D.G., L.S and F.C. All authors have read and agreed to the published version of the manuscript.

**Funding:** Dr. Lorenzo Muraglia is the recipient of a grant supported by the European Institute of Oncology Foundation (FIEO).

**Institutional Review Board Statement:** The study was conducted in accordance with the Declaration of Helsinki and approved by the local ethical committee and institutional scientific review board (IEO Trial-ID: 3490).

**Informed Consent Statement:** Informed consent was obtained from all subjects involved in the study.

**Data Availability Statement:** The authors confirm that the data supporting the findings of this study are available within the article. Supplementary data are available from the corresponding author, F.C., upon reasonable request.

**Conflicts of Interest:** The authors declare no conflict of interest regarding this study.

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
