# Peer review of "[18F]FDG PET/CT: Lung Nodule Evaluation in Patients Affected by Renal Cell Carcinoma"

_tomography, doi:10.3390/tomography9010031_

Round 1

Reviewer 1 Report

The clinical work from Farulla et al. exploited [18F]FDG-PET/CT to evaluate lung nodules in patients affected by renal cell carcinoma. The study is well organized with an acceptable inclusion criteria. Such studies are challenging in determining the tracer uptake specificity/sensitivity. More studies in different population subtypes are required for further validations. Following are the minor suggestions prior publishing in Tomography.   

1. Please add couple of PET images (whole-body) in the manuscript that show differences in FDG-uptake.

2. Suggest rephrasing the last sentence in abstract.. 'however......'

3. Is there a reason to acquire PET/CT images 60 minutes after the intravenous injection of the radiopharmaceutical?

4. Suggest rephrasing lines 230-237, as multiple sentences written together without period.

Author Response

Response to Reviewer-1

The clinical work from Airo’ Farulla et al. exploited [18F]FDG-PET/CT to evaluate lung nodules in patients affected by renal cell carcinoma. The study is well organized with an acceptable inclusion criteria. Such studies are challenging in determining the tracer uptake specificity/sensitivity. More studies in different population subtypes are required for further validations. Following are the minor suggestions prior publishing in Tomography.   

  1. Please add couple of PET images (whole-body) in the manuscript that show differences in FDG-uptake.

R: PET images have been added according to reviewer comment.

  1. Suggest rephrasing the last sentence in abstract.. 'however......'

R: The sentence has been amended according to reviewer comment.

  1. Is there a reason to acquire PET/CT images 60 minutes after the intravenous injection of the radiopharmaceutical?

R: All PET procedures were performed according to the European Association of Nuclear Medicine guidelines. These info have been disclosed in the text.

  1. Suggest rephrasing lines 230-237, as multiple sentences written together without period.

R: Sentences have been rephrased according to reviewer comment.

Reviewer 2 Report

Comments: The manuscript is well written and the study was well organised with good selection/ exclusion criteria.

The conclusions made are consistent with the data/results obtained and are very similar to those reported in other literature.

The investigation of whether  FDG uptake / TLG can differentiate RCC lesions from other lung metastases based on glycolysis is interesting, but the results are not unexpected.

The authors state that the data in this study was collected from 2004-2020, can the authors assure consistency in the imaging, collection of data and analysis of results over this large period?

The present study supports the results of related studies and in my opinion doing more/ larger cohorts may be of limited value.

In contrast, other radiopharmaceutical tracers should be evaluated such as FAPI

Author Response

Response to Reviewer-2

Comments: The manuscript is well written and the study was well organised with good selection/ exclusion criteria. The conclusions made are consistent with the data/results obtained and are very similar to those reported in other literature. The investigation of whether  FDG uptake / TLG can differentiate RCC lesions from other lung metastases based on glycolysis is interesting, but the results are not unexpected.

The authors state that the data in this study was collected from 2004-2020, can the authors assure consistency in the imaging, collection of data and analysis of results over this large period?

The present study supports the results of related studies and in my opinion doing more/ larger cohorts may be of limited value.

In contrast, other radiopharmaceutical tracers should be evaluated such as FAPI

R: We thank Reviewer-2 for the valuable comments. We support the reviewer statement regarding a larger patients cohort.  Of course, larger and multicentric studies might be needed to confirm this analysis. Probably other radiotracers (e.g. 89Zr-Girentuximab or FAP radiolabeled inhibitors) might be superior to detect RCC metastases. This comment has been implemented in the text. Regarding the further comment, we confirm that, out of the 350 patients evaluated from 2004 to 2020, we took the 70 patients who fulfilled all the inclusion/exclusion criteria, and the oldest scan was performed in 2010. We confirm that all procedures were performed according to international procedural guidelines (both considering injected activity or PET scan protocol), and not significant deviations were observed among different PET/CT scanners. Finally, we confirm that we are an EARL accredited center.

Reviewer 3 Report

Dear Author

This study highlighted the potential of discriminating metastatic lung nodules attributed to RCC on 18F FDG PET-CT.  You suggested that [18F]FDG-PET/CT to correctly identify lung metastases from RCC should be discouraged based on the low FDG affinity and small volume sampling. , as our analysis showed suboptimal diagnostic accuracy of [18F]FDG-PET/CT 283

I have few queiries on issues highlighted in the outcomes of the study as your suggestion that  [18F]FDG-PET/CT has shown to have added value, including prognostic value, setting of metastatic RCC patients especially  when the  volumetric parameters (MTV and TLG) are used to analyse.

1. Why there is no mention about primary RCC lesion on its dimension and characterisation and potentially a de-differentiated cells could habour altered glucose metabolism. Correlation between these parameters and the RCC lung nodules would give a better insight on the prediction of the histological landscape in the lung

2. Small volume lesion is below the PT resolution when the diameter is less than 1.0cm. I donts see any strong point you discussed on how the RCC micronodule would be discriminated from other nodules based on the cut-off SUV of 2.0.  You should state with cited evidences on how a qualitative value of SUV max of 2.0 can be discerned as those faint SUV activity below the mediastinal blood pool can be regarded as a negative PET

3. The MTV and TLG is just a firther extension of the SUV parameters which may not be valuable in characterising the glycolytic cellular derangement f a small nodule . I do not see how it could be added as a value to dstinguish from other nodules

4. instead of retrospectively use histology as a gold standard, immunohistopathology findings on genetic make up of the lesion, proliferative markers (ki 67)  are useful in dtermining the qualitative values of small volume lesions, based on the composite findings of structural (CT) and PET (funstional parameters) rather than a total reliance on the poor FDG activities.

4. It would be very useful to add relevant FDG PET images of the RCC lung nodules, primary RCC and non RCC lung nodules as a reference images.

Author Response

Response to Reviewer-3

This study highlighted the potential of discriminating metastatic lung nodules attributed to RCC on 18F FDG PET-CT.  You suggested that [18F]FDG-PET/CT to correctly identify lung metastases from RCC should be discouraged based on the low FDG affinity and small volume sampling. , as our analysis showed suboptimal diagnostic accuracy of [18F]FDG-PET/CT.

I have few queiries on issues highlighted in the outcomes of the study as your suggestion that  [18F]FDG-PET/CT has shown to have added value, including prognostic value, setting of metastatic RCC patients especially  when the  volumetric parameters (MTV and TLG) are used to analyze.

  1. Why there is no mention about primary RCC lesion on its dimension and characterisation and potentially a de-differentiated cells could habour altered glucose metabolism. Correlation between these parameters and the RCC lung nodules would give a better insight on the prediction of the histological landscape in the lung.

R: This is an interesting point and probably should deserve further analysis, that actually are outside the aim of the present study.  De-differentiation is difficult to analyze, and even if RCC is a fast-growing disease is still characterized by low FDG-avidity. Accordingly, it is not clear if higher FDG uptake might be expected by more de-differentiated RCC cell clones. In this study, we did not observe significant association between the tumor metabolism of the primary lesion and pulmonary lesion, namely considering that not all the patients analyzed received baseline FDG-PET to state the disease after RCC diagnosis.

  1. Small volume lesion is below the PET resolution when the diameter is less than 1.0cm. I donts see any strong point you discussed on how the RCC micronodule would be discriminated from other nodules based on the cut-off SUV of 2.0.  You should state with cited evidences on how a qualitative value of SUV max of 2.0 can be discerned as those faint SUV activity below the mediastinal blood pool can be regarded as a negative PET

R: We would thank the reviewer for this comment. The median diameter of the 111 lung nodules was 12 mm (IQR 8.5 – 18.5 mm). Of course, small dimensions and volumes might affect FDG-PET sensitivity. However, we observed that also bigger lesions had low FDG-uptake and, generally, FDG uptake was not a significant predictor of RCC vs. non-RCC lesions. Thus, although we agree with the reviewer about the importance of volume-dependent uptake in SUV derivates measurements, and even if non-RCC lesions had higher uptake, we were not able to define a statistically significant cut-off to distinguish between the two different tumor patterns.

  1. instead of retrospectively use histology as a gold standard, immunohistopathology findings on genetic make up of the lesion, proliferative markers (ki 67) are useful in dtermining the qualitative values of small volume lesions, based on the composite findings of structural (CT) and PET (funstional parameters) rather than a total reliance on the poor FDG activities.

R: We would thank reviewer regarding this brilliant comment. Unfortunately, ki67 and other genetic signatures were not available for all patients and, accordingly, this information were not included in our study. In further analysis enrolling greater cohorts (probably in a multicenter setting), these information will definitely add value to this sort of analysis.

  1. It would be very useful to add relevant FDG PET images of the RCC lung nodules, primary RCC and non RCC lung nodules as a reference images.

R: According to reviewer comment FDG-PET images have been added.

Round 2

Reviewer 3 Report

Dear author

The manuscript has been improved and it is good for publication